# Unraveling the optoelectronic properties of CoSb$_x$ intrinsic selective solar absorber towards high-temperature surfaces

Anastasiia Taranova [1], Kamran Akbar[1] ✉, Khabib Yusupov[2], Shujie You[3], Vincent Polewczyk[4], Silvia Mauri[4,5], Eleonora Balliana [6], Johanna Rosen[2], Paolo Moras[7], Alessandro Gradone [8], Vittorio Morandi [8], Elisa Moretti[1] ✉ & Alberto Vomiero [1,3] ✉

The combination of the ability to absorb most of the solar radiation and simultaneously suppress infrared re-radiation allows selective solar absorbers (SSAs) to maximize solar energy to heat conversion, which is critical to several advanced applications. The intrinsic spectral selective materials are rare in nature and only a few demonstrated complete solar absorption. Typically, intrinsic materials exhibit high performances when integrated into complex multilayered solar absorber systems due to their limited spectral selectivity and solar absorption. In this study, we propose CoSb$_x$ ($2 < x < 3$) as a new exceptionally efficient SSA. Here we demonstrate that the low bandgap nature of CoSb$_x$ endows broadband solar absorption (0.96) over the solar spectral range and simultaneous low emissivity (0.18) in the mid-infrared region, resulting in a remarkable intrinsic spectral solar selectivity of 5.3. Under 1 sun illumination, the heat concentrates on the surface of the CoSb$_x$ thin film, and an impressive temperature of 101.7 °C is reached, demonstrating the highest value among reported intrinsic SSAs. Furthermore, the CoSb$_x$ was tested for solar water evaporation achieving an evaporation rate of 1.4 kg m$^{-2}$ h$^{-1}$. This study could expand the use of narrow bandgap semiconductors as efficient intrinsic SSAs with high surface temperatures in solar applications.

The high surface temperature under solar light illumination is highly beneficial for a wide range of applications such as steam generation, passive heating, thermoelectricity generation, anti-icing/deicing, catalysis, converting a control signal into mechanical motion, and solar water evaporation[1–4]. Although surface temperature higher than 100 °C has been obtained under 1 sun illumination, this normally requires the

utilization of thermal insulation on top of the system to avoid heat losses[5]. This extra coverage on the top surface can restrict the penetrated light intensity due to its limited transparency and lead to condensation of generated vapors on the inner surface, commonly known as fogging. To overcome this drawback, various black absorber materials exhibiting broad solar absorption (0.3–2.5 μm) have been

[1]Department of Molecular Sciences and Nanosystems, Ca' Foscari University of Venice, Via Torino 155, 30172 Venice, Italy. [2]Department of Physics, Chemistry and Biology (IFM), Linköping University, 581 83 Linköping, Sweden. [3]Division of Materials Science, Department of Engineering Sciences and Mathematics, Luleå University of Technology, SE-971 87 Luleå, Sweden. [4]Istituto Officina dei Materiali (IOM) - CNR, Laboratorio TASC, Area Science Park, S.S. 14 Km 163.5, Trieste I-34149, Italy. [5]Dipartimento di Fisica, University of Trieste, via A. Valerio 2, 34127 Trieste, Italy. [6]Department of Environmental Sciences, Informatics and Statistics, Ca' Foscari University of Venice, Scientific Campus Via Torino 155/b, 30173 Venice, Italy. [7]Istituto di Struttura della Materia (ISM) - CNR, S.S. 14 Km 163.5, Trieste I-34149, Italy. [8]Istituto per la Microelettronica ed i Microsistemi (IMM) – CNR Sede di Bologna, via Gobetti 101, 40129 Bologna, Italy. ✉e-mail: kamran.akbar@unive.it; ELISA.MORETTI@unive.it; alberto.vomiero@ltu.se

developed, such as carbon and polymeric-based materials, but their surface temperatures remain below 85 °C owing to excessive thermal re-radiation back to the environment in the mid-infrared (IR) region (2.5–20 μm)[6,7]. This massive thermal re-radiation prevents heat concentration within the absorber and as a result, its surface temperature is drastically reduced. Following Kirchhoff's law of thermal radiation, for any material in thermal equilibrium, the ratio of the emissive power and the absorption coefficient is constant. This law has been applied to develop a promising class of materials called selective solar absorbers, for which a material with a low absorption coefficient in a specific spectral region will present low emission (and then low heat/energy dissipation) in the same region. SSAs are one of the classes of absorbers that can perfectly absorb the full-spectrum sunlight and avoid IR emission in the form of black body radiation, thanks to their low emission in the mid-IR region, avoiding thermal radiation heat losses. Hence, there is a dire need for materials possessing spectral selectivity with broad solar absorption ($\bar{\alpha}$) but simultaneous low mid-IR emission ($\bar{\varepsilon}$) (see Supplementary Fig. 1). The ability of a surface to absorb the light in the desired wavelength, while simultaneously reflecting the undesired wavelength is called spectral selectivity, which can be defined as the ratio between the average absorptance and emittance $\bar{\alpha}/\bar{\varepsilon}$.

Generally, in the reviews, SSAs surface coatings are classified into six categories depending on the structure and composition[8–10]. The first type is intrinsic SSA with desired intrinsic selective properties. The intrinsic SSA is the simplest in configuration among other types, though the major drawback is the scarcity of intrinsic SSAs in nature. Some of the transition metals and semiconductors (e.g., $TiO_2$, MgF, SrF) exhibiting some intrinsic selectivity properties need to be improved for practical use[11–15]. The second category includes composites of semiconductors and transition metals for attaining the desired spectral selectivity. In this combination, the low bandgap of the semiconductor is responsible for absorbing the short wavelengths, while the low emittance is provided by the metal layer[8]. The design of multilayer absorber solar selective coating consists of a substrate and multilayer dielectric (e.g., $Al_2O_3$, $SiO_2$, ZnS)/metal-metal oxide (e.g., Ag, Cu, Ni) combination and is based on the effect of multiple reflections between layers[16,17]. The combination of a metal substrate and absorber possesses good light absorption characteristics, while extinction interference absorption increases the solar absorption for the interface between substrate and absorber. To enhance light absorption, a smoother transition of the optical constant from the absorber to air is needed, which is provided by the dielectric layer that covers the absorber[18,19]. As multilayer solar selective coatings remain stable at a temperature above 400 °C, the further application can be attributed to the high-temperature range[20]. Cermet-based solar absorbers are the incorporated metal particles in a dielectric or ceramic matrix[21]. Owing to their excellent optical properties—a high absorptance under solar irradiation and high reflectance in the IR region—cermets have been extensively investigated (e.g., $Cr_2O_3$, $Al_2O_3$, $ZrO_2$ based)[22–24]. The selectively solar-transmitting coating on a blackbody-like absorber is commonly used for low-temperature applications (from ambient to 60 °C), while for high-temperature applications, the black absorber with highly doped semiconductors (e.g., $SnO_2$:Sb, $In_2SO_3$:Sn) has better performance[25–27]. The last concept of achieving desired spectral selectivity refers to the surface texturing of material. Thus, metals with smooth polished surfaces exhibit low emissivity, while metals with rough surfaces possess undesirably high emissivity. Additionally, the emissivity also depends on the temperature of the surface as well as on incidence and wavelengths.

Other strategies for achieving low intrinsic mid-IR emissivity and full absorption in the sunlight spectral range include various light manipulation techniques such as the fabrication of metasurfaces, photonic crystals, etc.[7]. In a recent study, Li et al. reported intrinsically low mid-IR emissivity (down to 0.1) but high and broad solar absorptance (up to 0.9) of two-dimensional (2D) $Ti_3C_2T_x$ MXenes thin films[6].

As a result, the thin film attained a surface temperature of 89 °C under 1 sun illumination. However, standalone materials possessing simultaneously high solar absorption and low mid-IR emission are scarce, and only a few have been reported, such as HfC, $ZrBr_2$, $CaF_2$, $SnO_2$, and $In_2O_3$, and their intrinsic spectral selectivity is still far from satisfactory[28–30].

In this study, the intrinsic spectral selectivity of 5.3 of semiconductor $CoSb_x$, ($2 < x < 3$, from this point forward, referred to as $CoSb_3$ from the main phase present in the sample) is demonstrated. Owing to its narrow bandgap of 0.26 eV, $CoSb_3$ exhibits a high and broad solar absorptance of 0.96 while its emissivity in the mid-IR region is only 0.18 at 100 °C. These are superior values compared to all intrinsic SSAs reported thus far. As a consequence, a thin film of $CoSb_3$ exhibits a record-high surface temperature of 101.7 °C among intrinsic SSA under 1 sun illumination. For proof of concept, the $CoSb_3$ was applied for solar water evaporation in a standard geometry using a glass fiber filter paper, which resulted in an evaporation rate of 1.4 kg m$^{-2}$ h$^{-1}$.

## Results
### Structural and morphology characterizations
The crystal structure of the $CoSb_3$ powder prepared by the solvothermal method (see "Methods" section) was investigated through different techniques. In X-ray Diffraction (XRD) measurements (see Supplementary Fig. 2a), the majority of the diffraction peaks confirmed the cubic skutterudite $CoSb_3$ phase (JCPDS #88–2437) of the *Im*-3 space group. A second phase is also present, and the diffraction peaks at 32.34°, 32.26°, 33.96°, and 50.78° can be attributed to the (111), (−121), (210), and (131) crystallographic planes of the monoclinic structure of $CoSb_2$ (JCPDS #29–126). The crystalline structures of $CoSb_3$ and $CoSb_2$ are presented in Supplementary Fig. 2b, c, respectively. Mi et al. reported that the formation of the $CoSb_3$ phase by solvothermal synthesis is a typical heat-activated reaction that is limited by the maximum treatment temperature of the autoclave during the synthesis[31]. An alternative approach to obtain the $CoSb_3$ phase is to increase the treatment time of the autoclave inside the oven, allowing the formation of single phase $CoSb_3$ by increasing the synthesis temperature up to 250 °C, while at lower temperatures the secondary phase $CoSb_2$ is formed as an intermediate product. Both $CoSb_3$ and $CoSb_2$ phases coexist in our Co-Sb system and present channels or voids available for the diffusion of water molecules, which suggests the potential application of Co–Sb composites for solar steam generation. Based on the Scherrer equation (Supplementary Note 1), the average crystallite size $D$ calculated from the strongest (013) reflection of the curve at 31.18° is 29.2 nm. From the Rietveld refinement analysis, the synthesized powder contains 72.8 wt.% $CoSb_3$ and 27.2 wt.% $CoSb_2$ (for details, see Supplementary Note 2).

Transmission Electron Microscopy (TEM) analysis (see Supplementary Fig. 3) gives the opportunity of a more detailed understanding of the system's structure. Supplementary Fig. 3a reports a TEM micrograph at low magnification of the cobalt antimonide sample where a particle-like shape with two different types of structures, enclosed in a μm-size aggregate is clearly shown in Fig. 1. The microstructure is composed of several particles with irregular shape, fused in larger structures or attached one to the other (blue dotted square) and a less concentrated secondary phase, with a rectangular shape (green dotted square). In Supplementary Fig. 3b, c are shown the high-resolution TEM (HRTEM) micrographs of the irregular particle, and of the one with rectangular shape, respectively. The former highlights the presence of several crystalline domains matching the one of $CoSb_3$ with a skutterudite cubic phase. The typical *d*-spacings of the skutterudite structure are highlighted in the FFT shown in the inset. The latter shows the presence of a different crystalline domain that corresponds to the one of a monoclinic $CoSb_2$ phase. The typical *d*-spacings of the monoclinic structure are highlighted in the FFT shown in

the inset. Finally, Supplementary Fig. 3d reports the selected area electron diffraction (SAED) pattern showing the typical behavior of a polycrystalline structure, due to the aggregation of many crystallites of different dimension and orientation. It is possible to recognize a polycrystalline pattern with the $d$-spacings of the cubic skutterudite phase ($CoSb_3$). It should also be noted the $d$-space value of 3.09 Å typical of a monoclinic phase $CoSb_2$. These results are in perfect agreement with the results obtained by XRD. The high-angle annular dark field (HAADF) scanning transmission electron microscopy (STEM) micrograph of the cobalt antimonide sample helps in the visualization of the irregular (highlight with red arrows) and rectangular (highlight with yellow arrow) particles, embedded in the micrometric-size aggregate.

Supplementary Fig. 3e, f show the HAADF-STEM image of the cobalt-based aggregate, and the regions (points marked with 2 and 3) where the energy-dispersive X-ray spectroscopy (EDS) measurements have been carried out and the resulting EDS spectra, respectively. The ratio in atomic percentage between Co and Sb changes, with a value of Co/Sb equal to 0.56 and 0.28 respectively for point 2 (rectangular particle) and point 3 (irregular particle), suggesting that the chemical composition and the shape are connected, in agreement with what observed by SAED and HRTEM analyses. The as obtained value perfectly match with the Co/Sb ratio respectively 0.5 of $CoSb_2$ and 0.33 of $CoSb_3$. This last observation confirms the coexistence of two different phases in the system.

Supplementary Fig. 4a, b shows the surface morphology of $CoSb_3$ composed of grains of slightly irregular shape, with a particle size of 15–40 nm with some agglomerations, which confirms the XRD diffraction and the Scherrer calculation. The porosity of the nanosized particles, well visible from the scanning electron microscopy (SEM) images, may favor water diffusion within the mesoporous layer, which forms when the membrane is fabricated. The corresponding EDS elemental mapping of the $CoSb_3$ sample (see Supplementary Fig. 4c, d) reveals the uniform distribution of Co and Sb over the film and gives a Co:Sb atomic ratio of 1:2.83, close to the expected value of 1:3. From the cross-sectional image of Supplementary Fig. 4e, the thickness of the $CoSb_3$ absorber layer after being deposited on the membrane for functional tests is estimated to be ~35 μm. X-ray photoelectron spectroscopy (XPS) measurements, presented in Supplementary Fig. 5, show the presence of three different Sb-related compounds, i.e. $CoSb_3$, $CoSb_x$ ($x < 3$), and $Sb_2O_3$ at the extreme surface of the sample. The detailed XPS analysis is reported in Supplementary Note 3.

## Optical measurements

Based on Fourier transform infrared (FTIR) data using the Tauc plot method presented in Fig. 2a it was established that $CoSb_3$ is a semiconductor with a direct band gap of ~0.26 eV[32]. This narrow bandgap allows interband electromagnetic radiation absorption to cover almost the entire solar spectrum. For this reason, $CoSb_3$ exhibits a high intrinsic absorptance ($\bar{\alpha}$) of 0.96 within the 0.3–2.3 μm wavelength spectral range (Fig. 2b), while simultaneously, the emissivity ($\bar{\varepsilon}$) is only 0.18. Based on these values, the spectral selectivity ($\bar{\alpha}/\bar{\varepsilon}$) of 5.3 is obtained, which is the highest value reported so far among intrinsic SSAs materials. The comparison between the current and reported record values is shown in Fig. 2c (see Supplementary Table 1).

## Density functional theory (DFT) calculations

To assess the mechanism behind the intrinsic spectral selectivity of the prepared sample, the optical properties of $CoSb_3$ and $CoSb_2$ were

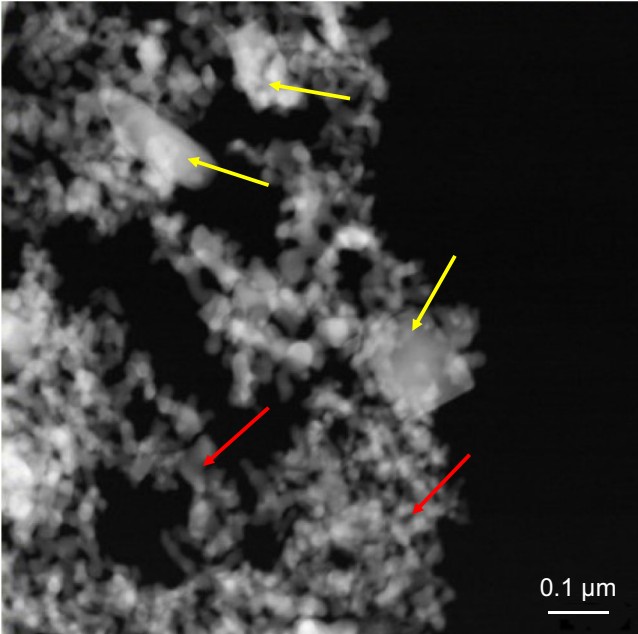

**Fig. 1 | HAADF−STEM micrograph of CoSb$_x$ sample.** The visualization of the irregular and rectangular particles, embedded in the micrometric-size aggregate, is highlighted with red and yellow arrows, respectively.

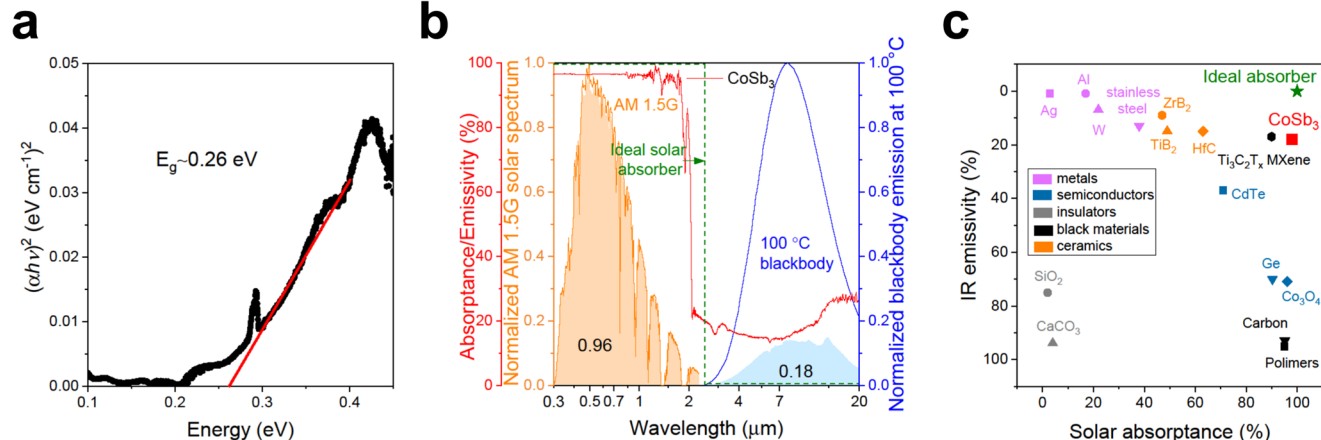

**Fig. 2 | Optical properties of CoSb$_3$. a** Tauc plot and calculation of band gap energy $E_g$ through the Tauc method. **b** Normalized absorptance/emissivity spectra of a $CoSb_3$ (red line), as well as the normalized AM 1.5 G solar spectrum (orange) and the normalized radiation spectrum of a blackbody at 100 °C (blue). The spectrum of an ideal absorber (green line). The light blue area indicates the emissivity of the present SSA, based on its experimentally measured absorptance. **c** Comparison of solar absorptance and IR emissivity of metals (Ag, Al, W, and stainless steel), radiative cooler materials ($SiO_2$ and $CaCO_3$), semiconductors (Ge, CdTe, and $Co_3O_4$), black materials (carbon-based and polymers), $TiB_2$, $ZrB_2$, HfC, and $Ti_3C_2T_x$ MXene.

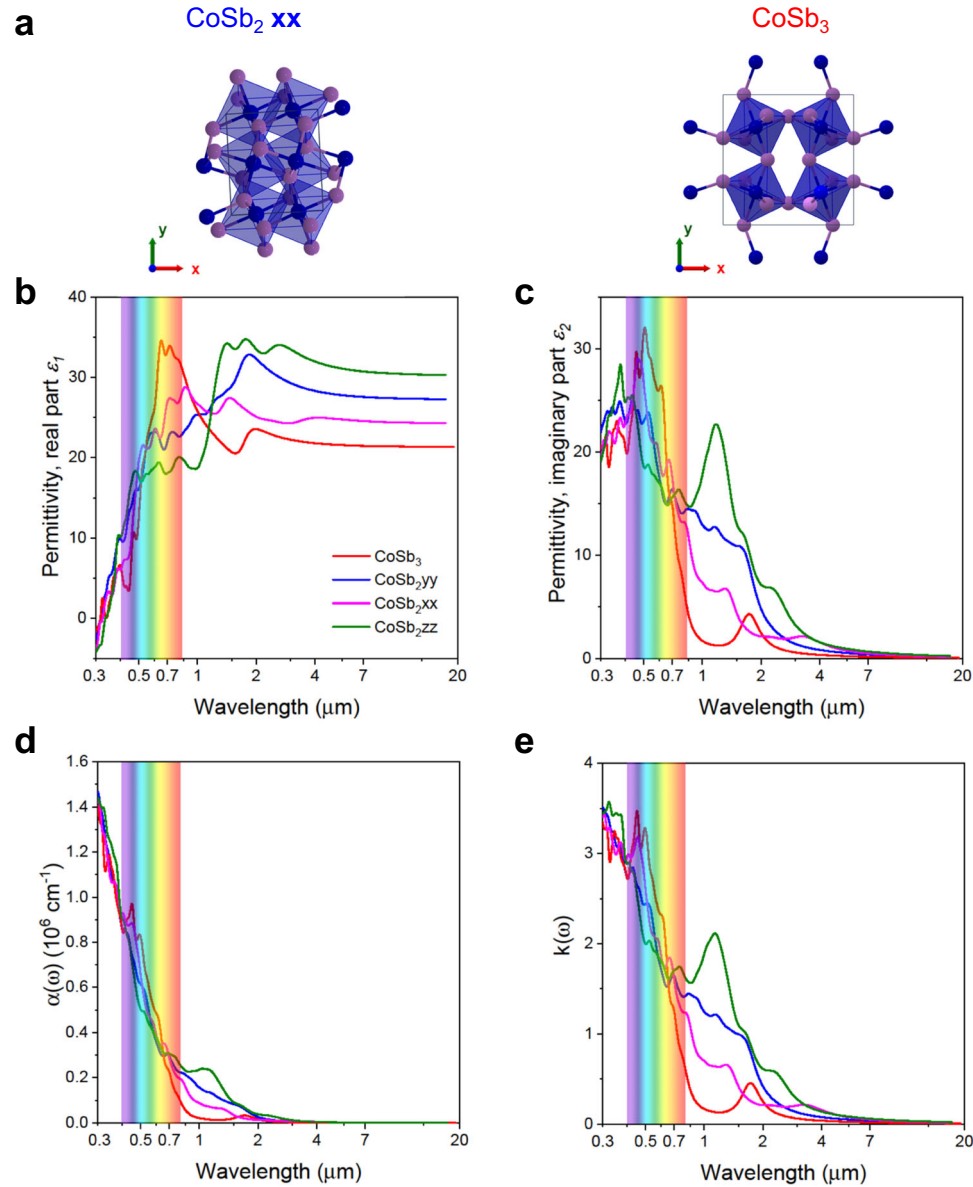

**Fig. 3 | Calculated optical properties of CoSb₃. a** Crystal structures of CoSb₂ and CoSb₃ in different directions (blue and purple balls represent Co and Sb respectively). **b, c** Real ($\varepsilon_1$) and imaginary ($\varepsilon_2$) part of permittivity. **d** The absorption coefficient of CoSb₂ and CoSb₃. **e** The extinction coefficient of CoSb₂ and CoSb₃.

evaluated by the DFT calculations (see Supplementary Note 4). The presented theoretical results should be treated separately and cannot be directly compared to the obtained in the laboratory results. Theoretical findings are used to investigate the contribution of various phases on the optical properties and not to provide the absolute values. They were obtained for the case where the crystal structure is infinite, boundaries are absent and other restrictions affecting the obtained results. Real-world systems deviate from these idealized conditions, and theoretical results reflect a qualitative trend rather than specific values. The optical properties of CoSb₂ are directionally dependent owing to its monoclinic structure and thus different results were obtained in the *xx*, *yy*, and *zz* directions, see schematic in Fig. 3a, while for CoSb₃, due to its cubic symmetry, calculated data from different directions displayed similar results upon interaction with incident electromagnetic radiation. The corresponding permittivity data consisting of real ($\varepsilon_1$) and imaginary ($\varepsilon_2$) parts are displayed in Fig. 3b, c, respectively. The $\varepsilon_1$ can be equated to the resistance of the system to the external field, i.e., the creation of an internal field as a response to the external one. In the region from ~0 to ~0.7 μm wavelength CoSb₃

exhibits growth of the permittivity (real part). Further on there is a decay, which is compensated by the directional behavior of CoSb₂ thus resulting in an almost constant built-in internal field. The $\varepsilon_2$ is associated with the absorption of the material, i.e., if the values are positive then the absorption is taking place. In the region from ~0.7 to ~2.0 μm there is a sudden drop in $\varepsilon_2$ for CoSb₃, which results in drop of absorption for that material. However, it is compensated by the imaginary part of CoSb₂ hence keeping the ability of the material to exhibit strong absorptance in the region from 0 to ~2 μm wavelength, which supports the experimental results and explains the mechanism behind the performance. Beyond wavelength values larger than 2 μm (mid-IR region), the $\varepsilon_2$ drops for both CoSb₃ and CoSb₂ suggesting low mid-IR emissivity. However, $\varepsilon_1$ values in this region remain constant after the initial buildup of internal field. This, combined with the simultaneous low mid-IR emissivity in the IR region, leads to higher surface temperature.

Furthermore, the calculated absorption coefficients (Fig. 3d) for both CoSb₃ and CoSb₂ can explain the strong absorption in the UV-vis-NIR range up to 1 μm while minimal absorption is observed in the mid-

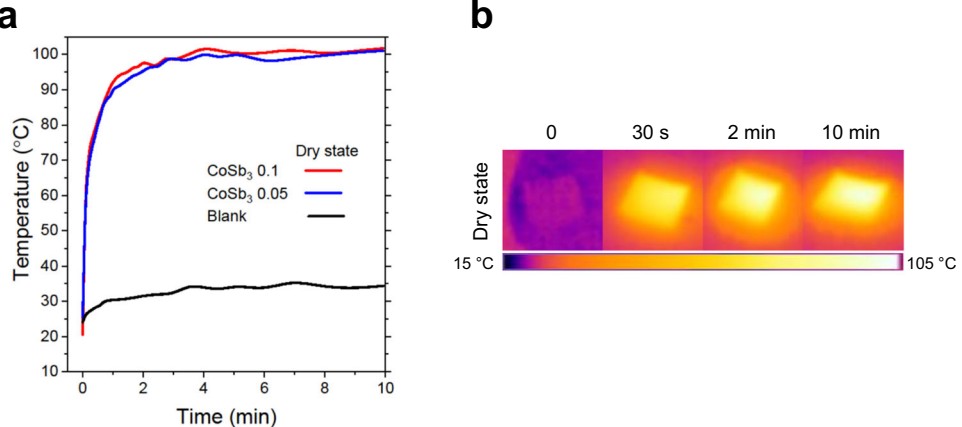

**Fig. 4 | Surface temperature of CoSb₃ at dry state under 1 sun. a** Photothermal behavior of wet CoSb₃ membrane under 1 sun illumination. **b** Time-lapse IR images of CoSb₃ 0.1 g membrane during water evaporation test.

IR region, strongly supporting the experimental observations. A similar trend has been observed for the extinction coefficient (Fig. 3e) of $CoSb_3$ and $CoSb_2$. Altogether, the DFT results indicate high absorption in UV-vis-NIR and low absorption in mid-IR. The optical features exhibited by $CoSb_3$ reproduce close to the optimal features for an ideal intrinsic SSA, i.e., very strong absorption within the 0.3–2.5 μm and almost negligible absorption for longer wavelengths. The calculated spectrum of transmittance and reflectance, normalized to unity, is shown in Supplementary Fig. 6a, b. From the reflectance spectrum the presence of $CoSb_2$ in the range 0.3–0.8 μm plays the dominant role in absorbing the incident light. The high theoretical reflectance values of around 0.5 and the high experimental average absorptance of 0.96 indicate that scattering is not negligible, and the secondary absorbance is taking place.

## Solar test

To estimate the surface temperatures under 1 sun illumination, the photothermal response of the powders in the membrane was assessed with a calibrated IR camera, as shown in Fig. 4a, b. Experiments were conducted at an ambient temperature (23 ± 3) °C and relative humidity ~50%. To determine the effect of the thickness of the absorber, two membranes with 0.05 and 0.1 g of loaded powder were tested. The dried thin film on the glass fiber membrane was placed on a Styrofoam insulator substrate and illuminated with AM (air mass) 1.5 G (global) solar irradiation (100 mW cm⁻² irradiance). After 30 sec of illumination, the temperature on the surface of $CoSb_3$ 0.1 rises from 25 °C to 82.0 °C. In 10 min of illumination, a saturated surface temperature of 101.7 °C was recorded at equilibrium. Benchmarking experiment with a non-SSA (carbon cloth, see description in the "Methods" section) resulted in maximum surface temperature of 63 °C at 1 sun, reached after about 5 min of continuous illumination. It is worth mentioning that achievement of this high surface temperature with $CoSb_3$ is observed without additional modifications such as surface texturing, addition of antireflection coatings, etc. Most of the solar absorbers, which are presented in Supplementary Table 2, demonstrates high surface temperatures greater than 100 °C. However, it is attributed to the utilization of multilayered structures (where SSA is sandwiched between antireflection coating and materials with low IR emissivity such as metals), thermal insulation, surface texturing, metamaterials etc. In addition, to date only a handful of intrinsic SSAs have been discovered thus far and when tested standalone under one sun irradiations, their surface temperatures remain lower than what we have observed for $CoSb_x$ thin films. The superior performance of 101.7 °C on the surface is the simultaneous combination of high intrinsic absorptance in the solar spectral region and low emissivity in the mid-IR region. High

intrinsic absorptance assures the maximum energy harvesting during irradiation, while simultaneous lower emissions in IR regions can suppress the heat loss to environment thus allowing heat concentration which ultimately leads to observed surface temperatures higher than 100 °C. The surface temperatures of $CoSb_3$ with 0.05 g loading of absorber material exhibit a saturation temperature of 100.9 °C, suggesting no effects from the thickness of the thin films. A blank test was performed to aid in the comparison. The result of the non-selective absorber confirms the role of low emissivity in the IR region for $CoSb_x$ film to achieve high surface temperature.

During the solar vapor generation test (see Supplementary Fig. 7 and Supplementary Note 5), the equilibrium temperature for $CoSb_3$ 0.05 g and $CoSb_3$ 0.1 g reached 48.9 °C and 50 °C, respectively, after 10 min. The equilibrium temperature for the carbon cloth was 38.7 °C. The $CoSb_3$ samples exhibited evaporation rates of 1.3 and 1.4 kg m⁻² h⁻¹, respectively, which are three times higher than the evaporation rate of bulk water under 1 sun illumination. The benchmarking carbon cloth exhibited a steady state evaporation rate equal to 1.08 kg m⁻² h⁻¹, which is significantly lower than the one obtained through the proposed SSA. This experimental evidence confirms once again the critical role of selective absorption to reduce the radiative heat losses and to maximize the evaporation rate.

According to the wettability test (see Supplementary Fig. 7e), the photothermal $CoSb_3$ membrane exhibits hydrophilic properties as a drop of water with a volume of 5 μL was absorbed within 696 ms.

Water exhibits strong absorption in the IR region. For this reason, wet samples during solar water evaporation may lose their spectral selectivity. We investigated the optical absorption properties in the 2–20 μm spectral range on dry and wet $CoSb_x$, to give a semi-quantitative estimate of the worsening of selective absorption/emission properties of the sample in the wet condition, compared to the dry state. The results are reported in Supplementary Note 6 and Supplementary Fig. 8. As clearly visible, the presence of water worsens the selective absorption, as expected, in the spectral range 2–20 μm. Still, despite the increased absorptance, the sample exhibits a clear selective absorption/emission in the critical spectral region for solar water evaporation. Under wet condition, the surface temperature decreases. However, spectral selectivity still holds, and it leads to a high evaporation rate, as experimentally reported.

In summary, a narrow bandgap solar absorber $CoSb_3$ has been synthesized by the solvothermal method and has been demonstrated as an ideal candidate for efficient photothermal conversion and highly efficient solar vapor generation under 1 sun illumination. The surface temperature of the $CoSb_3$ membrane under 1 sun in the open air achieved ~101.7 °C. This superior value was achieved by optimal

intrinsic optical properties—high $\bar{\alpha}$ of 0.96 across the solar spectrum and low $\bar{\varepsilon}$ of 0.18 in the mid-infrared region, achieving a record $\bar{\alpha}/\bar{\varepsilon}$ of 5.3. The results represent one of the best performances among all intrinsic SSAs ever reported possibly caused by the presence of $CoSb_2$, potentially improving the optical properties. The comparison with a benchmarking non-selective absorber indicates the importance to limit radiative heat losses to obtain a high surface temperature. Furthermore, the thin $CoSb_3$ membrane showed a high rate of water evaporation of ~1.4 kg m$^{-2}$ h$^{-1}$ under 1 sun illumination. The comparison with the benchmarking non-selective absorber shows that radiative emission in the IR region causes strong decrease of the evaporation rate, confirming that SSA is a viable strategy to obtain high evaporation rates. This study adds a new candidate to the existing small category of intrinsic SSA. The narrow bandgap and its association with high absorptance may open a field of research focused on narrow bandgap semiconductors possessing intrinsic broad solar absorptivity.

## Methods

### Synthesis

$CoSb_3$ was fabricated using the solvothermal synthesis. In a typical procedure, 3.6 mmol, 0.8213 g $SbCl_3$ (≥99.0%, Sigma-Aldrich; UN Hazard Class: 8; UN Pack Group: II) and 1.2 mmol, 0.2855 g $CoCl_2·6H_2O$ (98%, Sigma-Aldrich; Acute Tox. 4 Oral; Aquatic Acute 1; Aquatic Chronic 1; Carc. 1B Inhalation; Eye Dam. 1; Muta. 2; Repr. 1B; Resp. Sens. 1; Skin Sens. 1) were dissolved in a 120 mL ethanol bath. The solution was sonicated for 15 min, followed by the addition of 83.4 mol 0.4540 g $NaBH_4$, (≥98.0%%, Sigma-Aldrich; Acute Tox. 3 Oral; Eye Dam. 1; Repr. 1B; Skin Corr. 1B; Water-react 1) and sonicated for an additional 15 min for creating a reductive environment. Then the solution was sealed in a Teflon-lined stainless-steel autoclave with a 150 mL capacity and heated in an oven at 200 °C for 72 h. After cooling down to room temperature, the products were collected by centrifugation, washed using ethanol and deionized water several times, and dried at 70 °C overnight, isolated mass yields was estimated approximately 0.5 g.

The fabrication of the thin film membranes was performed through a vacuum filtration process. The powders with different masses of 0.05 and 0.1 g named $CoSb_3$ 0.05 and $CoSb_3$ 0.1, respectively, were dissolved in 250 mL of deionized water and then deposited on a glass microfiber filter paper with further drying.

### Materials characterizations

The crystalline structure of the samples was obtained using XRD on a Philips PW1050/37 diffractometer. Additional details on crystal structure, nanoscale morphology, and composition were obtained with a FEI Tecnai F20 High-Resolution Transmission Electron Microscope, equipped with a Schottky transmitter operating at 200 kV. The elemental analysis was carried out by EDS, coupled with STEM-HAADF to map the elemental distribution. SEM images were acquired by using a scanning electron microscope Carl Zeiss AG-SUPRA 40 equipped with EDS, Oxford Instruments. XPS was performed to determine chemical compositions in a dedicated chamber of the NFFA UHV MBE-cluster system[33,34]. The absorptance of the SSAs was measured using an ultraviolet-visible spectrophotometer LAMBDA 1050+, Perkin Elmer with an integrating sphere. A FTIR spectrometer (Nicolet 6700 FT-IR Spectrometer) equipped with a gold integrating sphere was used to record the IR spectrum of the compounds at room temperature.

The performance of a solar absorber can be evaluated by solar absorptance $\alpha$ and thermal emittance $\varepsilon$, which are calculated based on Kirchoff's law, spectrally averaged solar absorptance $\bar{\alpha}$, averaged thermal emissivity $\bar{\varepsilon}$ are described in detail in Supplementary Note 7. The measurement of solar reflectance by standard spectrophotometers is limited by the 0.3–2.3 μm wavelength range at near-normal $\theta = 0°$ angle of incidence.

The optical properties were calculated by the DFT using the hybrid Heyd–Scuseria–Ernzerhof (HSE) functional[35].

### Solar steam generation test in lab

The rapid photothermal response of membranes under 1 sun without bulk water was recorded using a solar simulator Abet Technologies model 10500 with simulated solar flux at 1 sun, calibrated with a Si standard solar cell. IR images of the membranes under solar illumination were taken by an IR camera (FLIR C3-X). The temperature output of the IR camera was calibrated by using a hotplate and a thermocouple integrated into the hotplate. Details on the calibration procedure are reported in Supplementary Note 8 and in Supplementary Figs. 9 and 10. The dry temperature measurements were performed by placing the membranes onto thermal insulator surfaces such as Styrofoam under 1 sun illumination. The solar test and solar vapor generation test were described in Supplementary Note 9 and Supplementary Fig. 11.

For benchmarking purposes, we considered the photothermal and evaporation properties of a standard, non-selectively absorbing material to compare with the proposed SSA. We selected a carbon cloth (from Phychemi. www.phychemi.com, n. WOS1009), whose composition was carbon for more than 99.9% atomic. The choice of a commercial product was dictated to guarantee the highest possible reproducibility of our results at independent labs. We measured both the maximum temperature under dry condition and the evaporation rate under simulated solar flux at 1 sun. The measurement conditions and the experimental set-up were the same as the ones used for the SSA $CoSb_3$.

The wettability of the membrane was tested by the measurement of the contact angle using Phoenix 300 Touch (SEO, South Korea) and the software Surfaceware9.

## Data availability

The authors declare that all data supporting the findings of this study are available within the article. The data generated in this study are provided in the Supplementary Information/Source data file. Source data are provided with this paper.

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

## Acknowledgements
A.V. acknowledges the Kempe Foundations, the Knut och Alice Wallenberg Foundation, the Luleå University of Technology (Labfund program) and the European Union - NextGenerationEU - through MUR (National Recovery and Resilience Plan (NRRP) – Mission 4 Component 2, Investment N. ECS00000043 – CUP N. H43C22000540006 (iNEST) and Mission 4 Component 2 Investment 1.3 - CUP D43C22003090001 (NEST)). This work has been carried out within the agreement "Convenzione operativa per collaborazione scientifica tra CNR ISM e Dipartimento di Scienze Molecolari e Nanosistemi Università Cà Foscari Venezia (Prot.n. 709, 14/04/2021)." Partial support through the project EUROFEL-ROADMAP ESFRI is gratefully acknowledged. J.R. acknowledges funding from the Göran Gustafsson foundation. The calculations were carried out using supercomputer resources provided by the Swedish National Infrastructure for Computing (SNIC), partially founded by the Swedish Research Council through grant agreement no. 2018-05973. This work has been partially performed in the framework of the Nanoscience Foundry and Fine Analysis (NFFA-MUR Italy Progetti Internazionali) facility (https://www.trieste.nffa.eu/). A.G. and V.M. acknowledge funding from the European Commission projects Graphene Flagship Core3, grant agreement No 881603, CHALLENGES, grant agreement No 861857, and from the European Union - NextGenerationEU under the National Recovery and Resilience Plan (NRRP), Mission 04 Component 2 Investment 3.1, Project Code: IR0000027 - CUP: B33C22000710006 - iENTRANCE@ENL: Infrastructure for Energy TRAnsition aNd Circular Economy @ EuroNanoLab

## Author contributions
A.T., K.A., E.M., and A.V. conceived the idea and designed the experiments. A.T., S.Y., K.A., and E.B. performed investigation on optical properties and solar water evaporation test. V.P., S.M., and P.M. performed the X-ray photoelectron spectroscopy measurements and data analysis. A.G. and V.M. performed HRTEM (EDS) and STEM-HAADF measurements and data analysis. K.Y. and J.R. conducted the density functional theory calculations and data analysis. A.T., K.A., and A.V. wrote the paper with contributions from all authors.

## Funding

## Competing interests
The authors declare no competing interests.
