## [Peer Review File · Nature Communications]

Unraveling the Optoelectronic Properties of CoSbx Intrinsic Selective Solar Absorber towards High-Temperature SurfacesREVIEWER COMMENTS

Reviewer #1 (Remarks to the Author):

The authors proposed an absorbing material (CoSbx) with high intrinsic spectral selectivity, studied the mechanism behind the optical properties by DFT calculation, and demonstrated its effectiveness in solar water evaporation. This material is innovative in the solar thermal application and the resulting optical properties are attractive, which may manifest a step forward to exploring intrinsic spectrally selective absorbers. However, for this high impact journal, the quality should be higher. I would recommend reconsideration after some major revisions.

1. The authors claimed “The low bandgap nature of CoSbx endows broadband solar absorption of 0.98 over the solar spectral range”, however, they did not provide the absorptance spectrum in the near-infrared band in Fig.1b. In fact, the transition from high solar absorptance to low thermal emittance plays a crucial role in solar absorptance. In other words, only a sharp transition can ensure a high solar absorptance of 0.98. In this situation, comparing the optical properties of CoSbx with other intrinsic absorbers may be not convincing, because the solar absorptance of many of those absorbers was measured over the solar spectral range, such as the Ti3C2Tx MXenes thin films.

2. The optical properties of the CoSbx absorber are indeed appealing. However, using the surface temperature increase as the main innovation point, especially as the title of this article, is inappropriate. The surface temperature increase is able to evaluate solar-thermal conversion capability. However, when radiated by sunlight, the surface temperature gain of an absorber is influenced by many factors. Besides the solar absorptance and thermal emittance of the absorber, environmental conditions, such as ambient temperatures and humidity, also significantly influence measurement results. Moreover, in this manuscript, the authors even did not provide environmental conditions during their surface temperature measurement. Also, the surface above 100 °C for a spectrally selective absorber is reported by many literatures, however, the authors claim “the surface temperature value of CoSb3 0.1 powder is the record-high”. As a result, the authors should clarify these points.

3. The authors employed an infrared camera to perform the surface temperature measurements. To the best of my knowledge, when an infrared camera is used to measure the surface temperature of a low emittance material, the calibration process by thermocouple or other routes is indispensable, otherwise, it may introduce some errors. Whether the measurement results are examined by thermocouple? I also note that in Fig. 3b surface temperature of the absorber is higher than its around temperature at $t = 0$. Considering the low emittance behavior of the absorber, more experimental details should be provided, such as materials underneath the absorber.

4. In Supplementary Note 5, the authors claim “This can be due to the high thermal conductivity of CoSb3 as now it can conduct generated heat rapidly to water and surrounding. This in turn inhibits heat localization, which is essential for a high evaporation rate.” This is an incorrect statement. The high thermal conductivity increases thermal loss to water and surrounding, which is detrimental to the

evaporation rate. In addition, heat localization is beneficial for evaporation rate. Researchers usually enhanced the evaporation rate by the increment of heat localization, which is able to inhibit thermal loss.

5. On page 3, the authors state “The top dielectric layer increases the absorption in the visible region”. However, in the spectrally-selective solar absorber, the top dielectric layer serves as an antireflection layer that decreases reflection loss on the absorber surface due to the sharp difference in refractive index between the air and absorbing layer (please refer to J. Mater. Chem. A, 2021, 9, 21270; Mater. Today Phys. 2022, 27, 100836). The top dielectric layer itself does not contribute to increasing solar absorption.

6. There also exist many careless mistakes. For example, in Supplementary Fig. 1. a, please delete “a” because only one Figure can be seen. In addition, in this Figure, “nir-IR” should be “near-IR”. In Supplementary Fig. 2, “(blue and purple balls represent Co and Sb respectively) should correspond to Supplementary Figure 2b,c rather than Supplementary Figure 2a. In Supplementary Note 7, “and the scheme of the experiment under 1 sun and dark conditions are presented in Supplementary Supplementary Fig. 6b, c”, please delete one of the “Supplementary”. On page 6, the authors state “Simultaneously, the emissivity is only 0.18 at a surface temperature of 25 °C”, however, on page 4, they also state “while its emissivity in the mid-IR region is only 0.18 at 100 °C”, what is the actual temperature of the emissivity of 0.18?

Reviewer #2 (Remarks to the Author):

The proposed intrinsic spectral selective absorber of CoSb₃ demonstrated a solar absorption of 98% (within 0.3-1.3 μm wavelength spectral range) and a thermal emissivity of 18% at 100 oC. The fabricated CoSb₃ thin film can obtain a surface temperature of ~101.7 oC under 1 kW m⁻² solar illumination and achieve a water evaporation rate of 1.4 kg m⁻² h⁻¹. However, the mechanism about the CoSb₃ selectivity and the reason for the 101.7 oC are not explained clearly. Some comments are listed here:

1. The authors mention that the CoSb₃ exhibits a high intrinsic absorptance of 0.98 within the 0.3-1.3 μm wavelength spectral range, and enables a record-high surface temperature of 101.7 oC under 1 kW m⁻² solar illumination. First of all, the manuscript didn't provide the absorption spectrum of CoSb₃ across the entire solar spectrum wavelength range. How about the entire solar spectrum weighted light absorption? And what is the main reason for the record-high surface temperature of 101.7 oC under 1 kW m⁻² solar illumination compared to early reported selective solar absorbers (the surface temperature can only reach ~85 oC under 1 kW m⁻² solar illumination in the open air conditions), such as TiN nanoparticle based absorber (α/ε = 95%/3% at 100 oC enables a higher light-to-thermal conversion efficiency than CoSb₃ absorber)[1]? Second, why do the authors believe that the 98% solar absorption within 0.3-1.3 μm wavelength range comes from the intrinsic absorption of CoSb₃? What's the contribution of the scattered effect of CoSb₃ nanoparticles on the total solar absorption? How to measure the surface temperature?

2. The authors calculate the permittivity of CoSb₃ and CoSb₂. However, the dielectric function of CoSb₂ seems unphysical for the negative imaginary part of permittivity, which means a negative refractive index or a negative extinction coefficient. Please give more explanations for this phenomenon.

3. This manuscript emphasizes the spectral selectivity of CoSb₃ solar absorber. However, the surface temperature of CoSb₃ thin film at wet state can only reach ~50 °C. Is the spectral selectivity maintained for CoSb₃ thin film at wet state? Or in another word, how much does the spectral selectivity of CoSb₃ thin film contribute to the water evaporation? Hence, introducing the contrast experiment of black absorber (without spectral selectivity) for water evaporation under the same conditions is more convincing.

4. In the manuscript, some statements are inconsistent. For example, on page 4, "CoSb₃ exhibits an emissivity of 0.18 at 100 °C", while at page 6, "the emissivity is 0.18 at a surface temperature of 25 °C". The authors should carefully check.

5. On page 7, "...from the mid-IR region (0.7 and 2 μm, respectively), ...", The mid-IR should be revised to near-IR. In figure 2b-e, the unit of horizontal title should be "μm".

References

[1] Y. Li, C. Lin, Z. Wu, Z. Chen, C. Chi, F. Cao, D. Mei, H. Yan, C.Y. Tso, C.Y.H. Chao, B. Huang, *Adv. Mater.* 2021, 33, 2005074.

Answer to REVIEWER COMMENTS (our answers in bold)

Reviewer #1 (Remarks to the Author):

The authors proposed an absorbing material (CoSbx) with high intrinsic spectral selectivity, studied the mechanism behind the optical properties by DFT calculation, and demonstrated its effectiveness in solar water evaporation. This material is innovative in the solar thermal application and the resulting optical properties are attractive, which may manifest a step forward to exploring intrinsic spectrally selective absorbers.

We are grateful to the reviewer for their encouraging words, for the recognition of the novelty of the proposed material and for prospective impact on intrinsic spectrally selective absorbers.

However, for this high impact journal, the quality should be higher. I would recommend reconsideration after some major revisions.

We recognize the need to improve the quality of the manuscript and we tried to clarify all the critical points raised by both the reviewers. In addition to the questions raised by the reviewers, we have added a systematic transmission electron microscopy (TEM) analysis to obtain insightful information on the local morphology, structure, and composition of the synthesized materials. In our opinion, the new measurements, in addition to the new calculations and calibration of temperature measurement system give a much more solid ground for the high functionality of the proposed materials.

1. The authors claimed “The low bandgap nature of CoSbx endows broadband solar absorption of 0.98 over the solar spectral range”, however, they did not provide the absorptance spectrum in the near-infrared band in Fig.1b. In fact, the transition from high solar absorptance to low thermal emittance plays a crucial role in solar absorptance. In other words, only a sharp transition can ensure a high solar absorptance of 0.98. In this situation, comparing the optical properties of CoSbx with other intrinsic absorbers may be not convincing, because the solar absorptance of many of those absorbers was measured over the solar spectral range, such as the Ti₃C₂T_x MXenes thin films.

Solar absorption was re-measured over a wider wavelength range of 0.3 – 2.3 μm. As can be seen by comparing the old and new Figure (reported below). According to the resulting plot, the solar absorptance remains high at 0.96 over the full absorption range. Although the spectral selectivity has changed from 5.4 to 5.3, this value is still high compared with other intrinsic absorbers.

Old (left) and new (right) figure reporting the absorptance of the sample in the range 0.3-20 μm . the missing window 1.2-2.3 μm has been filled with the new measurement campaign, enabling us to highlight the genuine nature of the SSA of our sample.

2. The optical properties of the CoSbx absorber are indeed appealing. However, using the surface temperature increase as the main innovation point, especially as the title of this article, is inappropriate. The surface temperature increase is able to evaluate solar-thermal conversion capability. However, when radiated by sunlight, the surface temperature gain of an absorber is influenced by many factors. Besides the solar absorptance and thermal emittance of the absorber, environmental conditions, such as ambient temperatures and humidity, also significantly influence measurement results. Moreover, in this manuscript, the authors even did not provide environmental conditions during their surface temperature measurement. Also, the surface above 100 °C for a spectrally selective absorber is reported by many literatures, however, the authors claim “the surface temperature value of CoSb3 0.1 powder is the record-high”. As a result, the authors should clarify these points.

We are grateful to the reviewer for recognizing the appealing optical properties of the newly proposed CoSbx absorber. We fully agree with their evaluation, and we count being able to further strengthen the manuscript to reach the scientific level acceptable for publication.

We thank the reviewer for their critical suggestion on surface temperature. We agree that surface temperature is only one of the parameters, which are involved in the process of selective solar absorption. We also agree that surface temperature increase enables the evaluation of solar-thermal conversion capability. This is the main motivation for us to stress the surface temperature in the title of the manuscript. However, following reviewer’s suggestion, we modified the title and the emphasis of the text, focusing more on the fundamental optical properties of our newly proposed selective solar absorber.

Capitalizing on reviewer's critical note, we have now given all the details about the environmental conditions during the measurements, such as ambient temperatures and humidity, which may significantly influence the results. Experiments were conducted at an ambient temperature and relative humidity around 50 %. We have also highlighted the importance of reporting the measurement conditions, which can be hardly found in the pertinent literature, affecting the reproducibility of the results in independent labs.

We apologize for our mistake that we did not specifically mention "intrinsic" spectrally selective absorber when we made the claim of record high temperatures by our samples. Selective absorber surface coatings can be categorized into six distinct types: a) intrinsic, b) semiconductor-metal tandems, c) multilayer absorbers, d) multi-dielectric composite coatings, e) textured surfaces, and f) selectively solar-transmitting coating on a blackbody-like absorber. Intrinsic absorbers use a material having intrinsic properties that result in the desired spectral selectivity.^[1] Only a handful of intrinsic spectrally selective absorber have been reported thus far and we have added a new member to intrinsic spectrally selective absorber family. We agree with reviewer that several publications exist, reporting surface temperature above 100 °C for a spectrally selective absorber. Supplementary Table 2 is now modified according to reviewer's comment. The achieved high value can be attributed to the modifications made with spectral selective absorber: texturing the surface, composite coatings with antireflection layer, multilayers coatings, etc. The critical statement in our manuscript is that CoSb_x possesses a high surface temperature among reported intrinsic spectrally selective absorbers when tested without any additional modifications.

We have altered the text as follows:

«It is worth mentioning that achievement of this high surface temperature with CoSb₃ is observed without additional modifications such as surface texturing, addition of antireflection coatings etc. The most of the solar absorbers, which are presented in Supplementary Table 2, demonstrates high surface temperatures greater than 100 °C, however it is attributed to the utilization of multilayered structures (where selective solar absorber is sandwiched between antireflection coating and materials with low IR emissivity such as metals), thermal insulation, surface texturing, metamaterials etc. In addition, to date only a handful of intrinsic selective solar absorbers have been discovered thus far and when tested standalone under one sun irradiations, their surface temperatures remain lower than what we have observed for CoSb_x thin films.»

1. C. Kennedy, *NREL Tech. Rep.* 2002, 1, <https://doi.org/10.2172/15000706>

3. The authors employed an infrared camera to perform the surface temperature measurements. To the best of my knowledge, when an infrared camera is used to measure the surface temperature of a low emittance material, the calibration process by thermocouple or other routes is

indispensable, otherwise, it may introduce some errors. Whether the measurement results are examined by thermocouple? I also note that in Fig. 3b surface temperature of the absorber is higher than its around temperature at $t = 0$. Considering the low emittance behavior of the absorber, more experimental details should be provided, such as materials underneath the absorber.

We thank the reviewer for their critical comments. We agree that an accurate calibration of the thermal camera is needed to provide a surface temperature measurement as precise as possible. For this reason, we carried out a calibration test, in which we measured the surface temperature of a black heater (Heidolph MR-HEI-TEC, 145 mm diameter) with the thermal camera, and simultaneously read the temperature of the heater measured by a thermocouple embedded in the heating plate. The results of the measurement are reported in the figure below, which we also included in the supporting information of the revised manuscript. We have selected a temperature range ($40\text{ }^{\circ}\text{C} < T < 110\text{ }^{\circ}\text{C}$) larger than the range at which our processes take place ($50\text{ }^{\circ}\text{C} < T < 100\text{ }^{\circ}\text{C}$), to have a good estimate of the validity of the temperature measurement.

As can be clearly seen in the figure below, there is a substantial agreement between the temperature measured by the thermocouple and the thermal camera. The linear fit has an intercept equal to $(-1.1 \pm 1.2)\text{ }^{\circ}\text{C}$ and a slope equal to $(1.035 \pm 0.015)\text{ }^{\circ}\text{C}/^{\circ}\text{C}$.

These results enable us to confirm that the temperature values given in the manuscript are highly accurate. However, to better identify potential uncertainties in the final value of the measured temperature, we have now set for each temperature an error, which accounts for the errors affecting the measurements.

Temperature measured through the thermal camera versus temperature measured by an internal thermocouple of a circular lab heater in the range 40-110 °C. The red solid line is the linear fitting of the experimental data.

4. In Supplementary Note 5, the authors claim “This can be due to the high thermal conductivity of CoSb₃ as now it can conduct generated heat rapidly to water and surrounding. This in turn inhibits heat localization, which is essential for a high evaporation rate.” This is an incorrect statement. The high thermal conductivity increases thermal loss to water and surrounding, which is detrimental to the evaporation rate. In addition, heat localization is beneficial for evaporation rate. Researchers usually enhanced the evaporation rate by the increment of heat localization, which is able to inhibit thermal loss.

We apologize for being unclear. We agree with the reviewer and that was the message we wanted to convey: high thermal conductivity is detrimental to the evaporation rate, since it increases thermal losses. For this reason, we rephrased as follows the text, highlighting that our target is to inhibit thermal losses.

We have altered the text as follows:

«For efficient water evaporation, minimizing heat loss is an important criterion that leads to maximizing the conversion efficiency. The absorber with lower thermal conductivity decreases heat loss, while CoSb₃ owes rather high thermal conductivity around 4.5 W·m⁻¹·K⁻¹. Nevertheless, the Styrofoam insulator substrate on which the absorber is placed prevents the heat losses to bulk water, minimizing heat transport.»

5. On page 3, the authors state “The top dielectric layer increases the absorption in the visible region”. However, in the spectrally-selective solar absorber, the top dielectric layer serves as an antireflection layer that decreases reflection loss on the absorber surface due to the sharp difference in refractive index between the air and absorbing layer (please refer to *J. Mater. Chem. A*, 2021, 9, 21270; *Mater. Today Phys.* 2022, 27, 100836). The top dielectric layer itself does not contribute to increasing solar absorption.

We agree with the reviewer: the top dielectric layer does not increase by itself solar absorption. It increases absorption “indirectly”, by minimizing the reflectance.

We have altered the text as follows:

«The combination of a metal substrate and absorber possesses good light absorption characteristics, while extinction interference absorption increases the solar absorption for the interface between substrate and absorber. To enhance light absorption, a smoother transition of the optical constant from the absorber to air is needed, which may be provided by the dielectric layer that covers the absorber^{21, 22}»

21 He, C. Y. *et al.* Toward high-temperature thermal tolerance in solar selective absorber coatings: choosing high entropy ceramic HfNbTaTiZrN. *J. Mater. Chem. A* 9, 21270–21280 (2021).

22. He, C. Y., Zhao, P., Gao, X. H., Liu, G. & La, P. Q. Enhancing thermal robustness of a high-entropy nitride based solar selective absorber by the incorporation of Al element. *Materials Today Physics* vol. 27 (2022).

6. There also exist many careless mistakes. For example, in Supplementary Fig. 1. a, please delete “a” because only one Figure can be seen. In addition, in this Figure, “nir-IR” should be “near-IR”. In Supplementary Fig. 2, “(blue and purple balls represent Co and Sb respectively) should correspond to Supplementary Figure 2b,c rather than Supplementary Figure 2a. In Supplementary Note 7, “and the scheme of the experiment under 1 sun and dark conditions are presented in Supplementary Supplementary Fig. 6b, c”, please delete one of the “Supplementary”. On page 6, the authors state “Simultaneously, the emissivity is only 0.18 at a surface temperature of 25 °C”, however, on page 4, they also state “while its emissivity in the mid-IR region is only 0.18 at 100 °C”, what is the actual temperature of the emissivity of 0.18?

We thank the reviewer for the notes. Changes have been made to the revised text. The actual temperature of the emissivity of 0.18 is 100 °C and we have clearly stated it in the revised manuscript.

Reviewer #2 (Remarks to the Author):

The proposed intrinsic spectral selective absorber of CoSb₃ demonstrated a solar absorption of 98% (within 0.3-1.3 μm wavelength spectral range) and a thermal emissivity of 18% at 100 oC. The fabricated CoSb₃ thin film can obtain a surface temperature of ~ 101.7 oC under 1 kW m⁻² solar illumination and achieve a water evaporation rate of 1.4 kg m⁻² h⁻¹. However, the mechanism about the CoSb₃ selectivity and the reason for the 101.7 oC are not explained clearly.

We thank the reviewer for the critical note. In addition to the questions raised by the reviewers, we have added a systematic transmission electron microscopy (TEM) analysis to obtain insightful information on the local morphology, structure, and composition of the synthesized materials. In our opinion, the new measurements, in addition to the new calculations and calibration of temperature measurement system give a much more solid ground for the high functionality of the proposed materials.

In the revised text we also tried to explain in a clearer and more complete way the mechanism behind selectivity, which results, as detailed below, from the reconsideration of the theoretical calculations of the optical properties of the newly proposed selective solar absorber.

“To assess the mechanism behind the intrinsic spectral selectivity of the prepared sample, the optical properties of CoSb₃ and CoSb₂ were evaluated by the density functional theory (DFT) calculations (see Supplementary Note 4). The optical properties of CoSb₂ are directionally dependent owing to its monoclinic structure and thus different results were obtained in the xx, yy and zz directions, see schematic in Error! Reference source not found.a, while for CoSb₃, due to its cubic symmetry, calculated data from different directions displayed similar results upon interaction with incident electromagnetic radiation. The corresponding permittivity data consisting of real (ϵ_1) and imaginary (ϵ_2) parts are displayed in Error! Reference source not found.b and c, respectively. The ϵ_1 can be equated to the resistance of the system to the external field, i.e., the creation of an internal field as a response to the external one. In the region from ~ 0 to ~ 0.7 μm wavelength CoSb₃ exhibits growth of the permittivity (real part). Further on there is a decay, which is compensated by the directional behavior of CoSb₂ thus resulting in an almost constant built-in internal field. The ϵ_2 is associated with the absorption of the material, i.e., if the values are positive then the absorption is taking place. In the region from ~ 0.7 to ~ 2.0 μm there is a sudden drop in ϵ_2 for CoSb₃, which results in drop of absorption for that material. However, it is compensated by the imaginary part of CoSb₂ hence keeping the ability of the material to exhibit strong absorptance in the region from 0 to ~ 2 μm wavelength, which supports the experimental results and explains the mechanism behind the performance. Beyond wavelength values greater than 2 μm (mid-IR region), the ϵ_2 drops for both CoSb₃ and CoSb₂ suggesting low mid-IR emissivity. However, ϵ_1 values in this region remain constant

after the initial buildup of internal field. When this is combined with simultaneous low mid-IR emissivity in this region, higher surface temperatures are expected.

Furthermore, the calculated absorption coefficients (Error! Reference source not found.d) for both CoSb_3 and CoSb_2 can explain the strong absorption in the UV-vis-NIR range up to $1 \mu\text{m}$ while minimal absorption is observed in the mid-IR region, strongly supporting the experimental observations. A similar trend has been observed for the extinction coefficient (Error! Reference source not found.e) of CoSb_3 and CoSb_2 . Altogether, the DFT results indicate high absorption in UV-vis-NIR and low absorption in mid-IR. The optical features exhibited by CoSb_3 reproduce close to the optimal features for an ideal intrinsic SSA, *i.e.* very strong absorption within the $0.3\text{--}2.5 \mu\text{m}$ and almost negligible absorption for longer wavelengths.”

Further text has also been added to revised version to better explain the reasoning for higher surface temperatures observed in this study.

“The superior performance of 101.7°C on the surface is the simultaneous combination of high intrinsic absorptance in the solar spectral region and low emissivity in the mid-IR region. High intrinsic absorptance assures the maximum energy harvesting during irradiation, while simultaneous lower emissions in IR regions can suppress the heat loss to environment thus allowing heat concentration which ultimately leads to observed surface temperatures higher than 100°C .

Some comments are listed here:

1. The authors mention that the CoSb_3 exhibits a high intrinsic absorptance of 0.98 within the $0.3\text{--}1.3 \mu\text{m}$ wavelength spectral range, and enables a record-high surface temperature of 101.7°C under 1 kW m^{-2} solar illumination. First of all, the manuscript didn't provide the absorption spectrum of CoSb_3 across the entire solar spectrum wavelength range. How about the entire solar spectrum weighted light absorption? And what is the main reason for the record-high surface temperature of 101.7°C under 1 kW m^{-2} solar illumination compared to early reported selective solar absorbers (the surface temperature can only reach $\sim 85^\circ\text{C}$ under 1 kW m^{-2} solar illumination in the open air conditions), such as TiN nanoparticle based absorber ($\alpha/\epsilon = 95\%/3\%$ at 100°C enables a higher light-to-thermal conversion efficiency than CoSb_3 absorber)[1]? Second, why do the authors believe that the 98% solar absorption within $0.3\text{--}1.3 \mu\text{m}$ wavelength range comes from the intrinsic absorption of CoSb_3 ? What's the contribution of the scattered effect of CoSb_3 nanoparticles on the total solar absorption? How to measure the surface temperature?

Solar absorption was re-measured over a wider wavelength range of $0.3\text{--}2.3 \mu\text{m}$. According to the resulting plot (see Figure below), the solar absorptance remains high at 0.96 over the full absorption range. Although the spectral selectivity has changed from 5.4 to 5.3, this value is still high compare with other intrinsic absorbers.

Old (left) and new (right) figure reporting the absorptance of the sample in the range 0.3-20 μm . the missing window 1.2-2.3 μm has been filled with the new measurement campaign, enabling us to highlight the genuine nature of the SSA of our sample.

We believe that the critical reason behind achieving this high surface temperature in comparison to other studies simultaneous buildup of internal field in response to external irradiation while maintaining low thermal emittance. In other studies, low thermal emittance is achieved via addition of metallic surface, metal do provide low mid-IR emissivity, however higher thermal conductivities of metals can result heat loss to surrounding and thus lower surface temperatures. Furthermore, the distinct metallic/other low mid-IR emissive layers provide external response i-e absorption is taking place on separate layer while low emissivity is provided elsewhere. Although, thermal response can be cumulative but still thermal losses in such multilayer structures cannot be ignored. Thus, there is a dire need for development of standalone materials which provide both high absorption and low mid-IR emissivity.

Based on the results of morphological and structural characterization, we think we can rule out the effect of scattering on absorption properties of our material. The lateral dimensions of our particles are much lower than the involved wavelengths in the visible range, so we expect a negligible role of scattering, while the absorption properties can be satisfactorily understood in terms of the intrinsic optoelectronic properties of the proposed material.

The surface temperature was recorded with an IR camera (FLIR C3-X) and the calibration for surface temperature measurements was described in the answer to comments from reviewer #1.

2. The authors calculate the permittivity of CoSb3 and CoSb2. However, the dielectric function of CoSb2 seems unphysical for the negative imaginary part of permittivity, which means a negative

refractive index or a negative extinction coefficient. Please give more explanations for this phenomenon.

We are grateful to the reviewer for pointing out this issue. The negative values of permittivity have previously been reported by Javan *et al.* [DOI:10.1007/s11468-018-0795-2]. The authors found this behavior in the interband dielectric constant for the case when the electric field is perpendicular to the basal plane of bulk graphite. However, since the YZ direction provides no information relevant to the present study, we have reconsidered our choice of directions (XX, YY, etc.) to be evaluated, and focus on those, which are relevant and which allow comparison to experimental results. Hence, we prioritized XX, YY, and ZZ directions, since the analysis of the literature shows that these are the planes that are mostly used for evaluation of the imaginary part of the permittivity. We have updated the text and the figure in the paper:

“The optical properties of CoSb₂ are directionally dependent owing to its monoclinic structure and thus different results were obtained in the xx, yy and zz directions, see schematic in Error! Reference source not found.a, while for CoSb₃, due to its cubic symmetry, calculated data from different directions displayed similar results upon interaction with incident electromagnetic radiation. The corresponding permittivity data consisting of real (ϵ_1) and imaginary (ϵ_2) parts are displayed in Error! Reference source not found.b and c, respectively. The ϵ_1 can be equated to the resistance of the system to the external field, i.e., the creation of an internal field as a response to the external. In the region from ~ 0 to $\sim 0.7 \mu\text{m}$ wavelength CoSb₃ exhibits growth of the permittivity (real part). Further on there is a decay, which is compensated by the directional behavior of CoSb₂ thus resulting in an almost constant built-in internal field. The ϵ_2 is associated with the absorption of the material, i.e., if the values are positive then the absorption is taking place. In the region from ~ 0.7 to $\sim 2.0 \mu\text{m}$ there is a sudden drop in ϵ_2 for CoSb₃, which results in drop of absorption for that material. However, it is compensated by the imaginary part of CoSb₂ hence keeping the ability of the material to exhibit strong absorptance in the region from 0 to $\sim 2 \mu\text{m}$ wavelength, which supports the experimental results and explains the mechanism behind the performance. Beyond wavelength values greater than $2 \mu\text{m}$ (mid-IR region), the ϵ_2 drops for both CoSb₃ and CoSb₂ suggesting low mid-IR emissivity. However, ϵ_1 values in this region remain constant after the initial buildup of internal field, when this combined with simultaneous low mid-IR emissivity in this region, higher surface temperatures are expected.”

3. This manuscript emphasizes the spectral selectivity of CoSb_3 solar absorber. However, the surface temperature of CoSb_3 thin film at wet state can only reach $\sim 50^\circ\text{C}$. Is the spectral selectivity maintained for CoSb_3 thin film at wet state? Or in another word, how much does the spectral selectivity of CoSb_3 thin film contribute to the water evaporation? Hence, introducing the contrast experiment of black absorber (without spectral selectivity) for water evaporation under the same conditions is more convincing.

Owing no emission in the IR region, an ideal solar absorber inhibits radiative heat losses leading to high solar-thermal efficiency. Therefore, the spectral selectivity provides the absorber to generate steam under 1 sun illumination even for applications with high temperature of steam

(water boiling, sterilization). In our work we did not investigate the contribution of the spectral selectivity to the evaporation rate, while Yin et al. (1) compared the ability to efficiently generate solar evaporation of selective ($\text{Sm}_{0.5}\text{Sr}_{0.5}\text{CoO}_{3-\delta}$) and nonselective ($\text{La}_{0.5}\text{Sr}_{0.5}\text{Co}_{0.5}\text{Ni}_{0.5}\text{O}_{3-\delta}$) perovskite oxide. Compared to nonselective absorber, SSA was more conducive to contributing to solar evaporation and revealed higher evaporation performance, which was increased by ~9% due to lower thermal radiation loss. The computed heat fluxes showed that radiative heat flux of SSA was lower than nonselective absorber, while the evaporative heat flux of SSA was possessing significantly higher values than for the non-SSA. The reason for the higher evaporation rate of SSA was linked to the generated heat in SSA, which was converted into evaporation enthalpy of bulk water. The faster steam generation reduced air convection heat loss on the surface, and the faster upward transport of water reduced conduction heat loss to the bulk water. Based on that research, we can claim that spectral selectivity of CoSb_3 significantly contributes to its evaporation rate.

In case of blackbody-like absorber the harvested solar energy will be mostly eliminated because of the thermal re-radiation. Unfortunately, the information about solar evaporation test with blackbody material ($\alpha/\varepsilon=1$) was not found in the literature. A near-perfect blackbody absorber (for example, carbon nanotubes CNT) with applying texturing or designing a multilayer absorber structure has been used as SSA. A CNTs absorber shows the surface temperature 50 °C and spectral selectivity $\alpha/\varepsilon=1.02$ under 1 sun irradiance, leading to a solar-thermal efficiency $\eta_{\text{solar-th of}} \approx 32\%$. The evaporation rate for a CNT-based floating solar evaporator is still $0.88 \text{ kg}\cdot\text{m}^{-2}\cdot\text{h}^{-1}$, under 1 sun illumination (2).

(1) Yin, J. *et al.* Selective Ceramic Absorber with Vertical Pore Structure for Efficient Solar Evaporation. *Sep. Purif. Technol.* 2022, 292 (March), 121009. <https://doi.org/10.1016/j.seppur.2022.121009>.

(2) Gan, Q.; Zhang, T.; Chen, R.; Wang, X.; Ye, M. Simple, Low-Dose, Durable, and Carbon-Nanotube-Based Floating Solar Still for Efficient Desalination and Purification. *ACS Sustain. Chem. Eng.* 2019, 7 (4), 3925–3932. <https://doi.org/10.1021/acssuschemeng.8b05036>

4. In the manuscript, some statements are inconsistent. For example, on page 4, “ CoSb_3 exhibits an emissivity of 0.18 at 100 °C”, while at page 6, “the emissivity is 0.18 at a surface temperature of 25 °C”. The authors should carefully check.

We apologize for the clerical error. An emission value of 0.18 is correct at a temperature value of 100 °C.

5. On page 7, “...from the mid-IR region (0.7 and 2 μm , respectively), ...”, The mid-IR should be revised to near-IR. In figure 2b-e, the unit of horizontal title should be “ μm ”.

We thank the reviewer for the note. Changes have been made to the revised text.

REVIEWER COMMENTS

Reviewer #1 (Remarks to the Author):

I carefully read the author's response to reviewer comments and the revised version of the paper "Unraveling the Optoelectronic Properties of CoSbx Intrinsic Selective Solar Absorber towards High-Temperature Surfaces". Most of the reviewer's concerns have been addressed and the paper has improved. However, there are still some concerns regarding solar water evaporation that need to be addressed.

The authors show that the CoSbx SSA has a high thermal conductivity of around $4.5 \text{ W}\cdot\text{m}^{-1}\cdot\text{K}^{-1}$, which inevitably results in conduction loss with its surroundings, even with the use of heat insulation material to reduce this loss. Nevertheless, the evaporation rate of $1.4 \text{ kg}\cdot\text{m}^{-2}\cdot\text{h}^{-1}$ is almost close to the upper limit of $1.47 \text{ kg}\cdot\text{m}^{-2}\cdot\text{h}^{-1}$ for two-dimensional evaporators. The authors attribute this high evaporation rate to the intrinsic spectral selectivity of the CoSbx absorber, which is believed to inhibit radiative thermal loss due to its low thermal emittance. However, as mentioned by Reviewer #2, the CoSb₃ thin film may lose its spectral selectivity at the wet state due to the intrinsic high emittance of water in the IR region. Have the authors attempted to verify this point? Given the above concerns, it may be necessary to explore further mechanisms or steady-state thermal equilibrium to support the evaporation rate.

Reviewer #2 (Remarks to the Author):

The current version of the manuscript still has a big step. Some questions need to be further addressed as follows.

1. The absorptance profile shows a drastic drop ($\sim 70\%$) between $2.3 \mu\text{m}$ and $2.5 \mu\text{m}$, which is highly favorable to selective spectral absorption. Hence, it is recommended to measure the absorptance spectrum of the absorber in this wavelength range.
2. Authors state that the attribution of scattering on total absorption of the absorber is negligible due to the smaller lateral dimension of the particle than the incident wavelength. However, according to the results of the calculated permittivity of the material, the refractive index is around 5 within the visible wavelength, which can result in a big reflectance in air due to the mismatched refractive index. Please give more explanations.
3. According to the calculated permittivity of the material, validity is recommended to provide through simulating the reflectance and transmittance of materials using optical permittivity.
4. The spectral selectivity is favorable for water evaporation. However, during the working process of

water evaporation, the water or water steam would appear inside the absorber, influencing the spectra selectivity. Hence, it is crucial to confirm whether the spectral selectivity of CoSb₃ film is still maintained in a wet state. Therefore, providing the emission/absorption spectra of CoSb₃ films in a wet state is recommended. In addition, the comparison of the evaporation rate for CoSb₃ film and black absorber (without spectral selectivity) should be given at the same experimental condition.

ANSWERS TO REVIEWER COMMENTS (our answers in bold)

Reviewer #1 (Remarks to the Author):

I carefully read the author's response to reviewer comments and the revised version of the paper "Unraveling the Optoelectronic Properties of CoSbx Intrinsic Selective Solar Absorber towards High-Temperature Surfaces". Most of the reviewer's concerns have been addressed and the paper has improved.

We are glad to read that the reviewer is favorably evaluating our revised manuscript, recognizing that most of the reviewer's concerns have been addressed and the paper has improved.

However, there are still some concerns regarding solar water evaporation that need to be addressed.

We thank the reviewer for the critical note. In the revised manuscript we have tried our best to address all the remaining criticisms.

The authors show that the CoSbx SSA has a high thermal conductivity of around $4.5 \text{ W}\cdot\text{m}^{-1}\cdot\text{K}^{-1}$, which inevitably results in conduction loss with its surroundings, even with the use of heat insulation material to reduce this loss. Nevertheless, the evaporation rate of $1.4 \text{ kg}\cdot\text{m}^{-2}\cdot\text{h}^{-1}$ is almost close to the upper limit of $1.47 \text{ kg}\cdot\text{m}^{-2}\cdot\text{h}^{-1}$ for two-dimensional evaporators.

We apologize with the reviewer for the confusion. We reported the value of thermal conductivity for CoSb_x ($4.5 \text{ W m}^{-1} \text{ K}^{-1}$) from the literature. We did not measure it in our sample. We clarified this point in the revised manuscript. We agree with the reviewer that a high thermal conductivity may affect in heat losses due to conduction. Still, the presence of the heat insulating material at the bottom of the CoSb_x film seems being effective in mitigating heat losses, enabling to reach a high evaporation rate.

The authors attribute this high evaporation rate to the intrinsic spectral selectivity of the CoSbx absorber, which is believed to inhibit radiative thermal loss due to its low thermal emittance. However, as mentioned by Reviewer #2, the CoSb₃ thin film may lose its spectral selectivity at the wet state due to the intrinsic high emittance of water in the IR region. Have the authors attempted to verify this point?

We thank both the reviewers for the observation about the intrinsic spectral selectivity and evaporation rate under dry and wet conditions. For sure the presence of water and its broadband

absorption worsens the selective absorption/emission properties of the absorber, increasing the heat loss due to radiative emission in the IR region. The question about the use of CoSb_x as selective solar absorber is to be able to quantify the worsening of selective absorption/emission properties of the sample in the wet condition, compared to the dry state. For this reason, we report below the absorptance properties of the CoSb_x sample under dry and wet conditions, together with the blackbody emission at 100 °C. As clearly visible, the presence of water worsens the selective absorption, as expected. Still, despite the increased absorptance, the sample exhibits a clear selective absorption/emission in the critical spectral region for solar water evaporation. Of course, under wet condition, the surface temperature decreases, as reported in our study. However, spectral selectivity still holds, and, in our opinion, it leads to a high evaporation rate, as experimentally reported.

We have added this information in the revised supporting information.

Given the above concerns, it may be necessary to explore further mechanisms or steady-state thermal equilibrium to support the evaporation rate.

As per reviewer's suggestion, we have tried to demonstrate and further validate the evaporation rate obtained by our proposed SSA. We benefited also from suggestion of reviewer #2, indicating the possible use of a benchmarking non-SSA material to compare our results.

As per reviewers' suggestion, we have investigated the functional properties of a black absorber without spectral selectivity. To allow a fair and easy comparison of the results and reproduce them at independent labs, we have chosen as black absorber a carbon cloth (product number W0S1011 from PHYCHEMi, www.phychemi.com). We have carried out systematic characterization of the carbon cloth in terms of both surface temperature in dry and wet conditions and we measured the

evaporation rate under the same experimental conditions. We applied carbon cloth in the same geometry as the CoSb_x . We first investigated the asymptotic temperature achieved by carbon cloth in dry conditions under one sun illumination. The asymptotic temperature was reached by the carbon cloth sample after 7 minutes of continuous irradiation. A final temperature of 63 °C was achieved by the sample, which is far below the final temperature obtained by the CoSb_x sample (~101.7 °C). This is a rather convincing argument supporting the choice of CoSb_x not only for water evaporation, but also for other high-surface temperature applications, where high surface temperature is targeted. After having compared the maximum temperature achieved in dry conditions, we have measured the evaporation rate from the carbon cloth sample under the same experimental conditions applied for the CoSb_x system. The results of the evaporation rate are reported in the figure below and are added in the Supporting information section of the revised manuscript.

Evaporation efficiency of the membranes under 1 sun irradiation and dark condition. The evaporation rate obtained by applying a benchmarking carbon cloth is also reported.

As it is clearly visible, after 20-minute transient, the results on water evaporation from the carbon cloth (1.08 kg m⁻² h⁻¹) are systematically lower than the ones obtained from the CoSb_x system, indicating that CoSb_x guarantees an improved mechanism of heat management during water evaporation, compared to a black, non-selective absorber.

To further elucidate the functional features of the proposed SSA contributing to the high evaporation rate, we carried out wettability tests, reported as the new Supplementary Figure 6e (also reported

below). According to the wettability test (see Supplementary Fig.7e) the photothermal CoSb_3 membrane exhibits hydrophilic properties as a drop of water with a volume of $5 \mu\text{L}$ was absorbed within 696 ms.

Time-lapse snapshots of absorbing a water droplet by the photothermal CoSb_3 membranes.

We are grateful to reviewers for having suggested this control experiment, which further clarifies the originality and significance of our work.

We are also confident that the new data can solidify the conclusions of our work, obtaining a clear demonstration of the advantages offered by our new SSA as simple system for high-surface temperature and high solar water evaporation.

Reviewer #2 (Remarks to the Author):

The current version of the manuscript still has a big step. Some questions need to be further addressed as follows.

1. The absorptance profile shows a drastic drop ($\sim 70\%$) between $2.3\ \mu\text{m}$ and $2.5\ \mu\text{m}$, which is highly favorable to selective spectral absorption. Hence, it is recommended to measure the absorptance spectrum of the absorber in this wavelength range.

We thank the reviewer for the suggestion. Indeed, the discontinuity in the measurement range, which affected the previously reported data, represents a weakness from the viewpoint of demonstrating the selective spectral absorption properties of the proposed semiconductor. The main issue to obtain a continuous spectrum covering the full range, especially in the region $1\text{-}3\ \mu\text{m}$ is due to the lack of overlap of the available range for the UV-vis spectrometer and the FT-IR spectrometer. In fact, in the previous measurements, we were unable to cover the range down to $2\ \mu\text{m}$ with our FT-IR (low energy side) and the high energy side was characterized by a full absorption up to $2.3\ \mu\text{m}$. However, the signal coming from the UV-vis spectrometer was very noisy in the low energy side and it was saturating the absorption signal mainly due to the extremely high thickness of the measured sample. We worked on the preparation of the sample, and we were able to extend the range of the UV-vis spectrometer and to obtain two distinct signals from the UV-vis and FT-IR spectrometers, overlapping in the region around $2\ \mu\text{m}$ (see image below).

These new measurements indicate that the proposed semiconductor is genuinely a spectrally selective absorber, whose absorption onset falls exactly in the region suitable to absorb the full solar

spectrum, avoiding radiative heat loss at high temperatures. We were unable to remove the slight discontinuity, which is present at around 2 μm , which is the region where the switch between the UV-vis and the FT-IR spectrometers falls.

We have reported these new measurements in the revised manuscript.

2. Authors state that the attribution of scattering on total absorption of the absorber is negligible due to the smaller lateral dimension of the particle than the incident wavelength. However, according to the results of the calculated permittivity of the material, the refractive index is around 5 within the visible wavelength, which can result in a big reflectance in air due to the mismatched refractive index. Please give more explanations.

Please see the response to comment №3

3. According to the calculated permittivity of the material, validity is recommended to provide through simulating the reflectance and transmittance of materials using optical permittivity.

We very much appreciate the helpful comment and opportunity to present an additional analysis. From the calculated and normalized transmittance and reflectance graph presented below and relation $A=1-R-T$ the absorptance for DFT-calculated CoSb_3 and CoSb_2 in the solar spectrum cannot reach more than ~ 0.50 . From the reflectance spectrum the presence of CoSb_2 in the range 0.3-0.8 μm plays the dominant role in absorbing the incident light. Theoretical findings are used to investigate the contribution to various phases on the optical properties and not to provide the absolute values. They were obtained for the case where the crystal structure is infinite, boundaries are absent and other real-world restrictions affecting the obtained results are not present. As the reflectance and scattering are interconnected phenomena, and according to the high values of calculated reflectance, the effect of scattering is not negligible and secondary absorbance is taking place. The latter results in high experimental absorptance 0.96.

We have added this information in the revised supporting information.

4. The spectral selectivity is favorable for water evaporation. However, during the working process of water evaporation, the water or water steam would appear inside the absorber, influencing the spectra selectivity. Hence, it is crucial to confirm whether the spectral selectivity of CoSb₃ film is still maintained in a wet state. Therefore, providing the emission/absorption spectra of CoSb₃ films in a wet state is recommended. In addition, the comparison of the evaporation rate for CoSb₃ film and black absorber (without spectral selectivity) should be given at the same experimental condition.

We thank both the reviewers for the observation about the intrinsic spectral selectivity and evaporation rate under dry and wet conditions. For sure the presence of water and its broadband absorption worsens the selective absorption/emission properties of the absorber, increasing the heat loss due to radiative emission in the IR region. The question about the use of CoSb_x as selective solar absorber is to be able to quantify the worsening of selective absorption/emission properties of the sample in the wet condition, compared to the dry state. For this reason, we report below the absorptance properties of the CoSb_x sample under dry and wet conditions, together with the blackbody emission at 100 °C. As clearly visible, the presence of water worsens the selective absorption, as expected. Still, despite the increased absorptance, the sample exhibits a clear selective absorption/emission in the critical spectral region for solar water evaporation. Of course, under wet condition, the surface temperature decreases, as reported in our study. However, spectral selectivity still holds, and, in our opinion, it leads to a high evaporation rate, as experimentally reported.

We have added this information in the revised supporting information.

As per reviewer's suggestion, we have tried to demonstrate and further validate the evaporation rate obtained by our proposed SSA. We benefited also from suggestion of reviewer #2, indicating the possible use of a benchmarking non-SSA material to compare our results.

As per reviewers' suggestion, we have investigated the functional properties of a black absorber without spectral selectivity. To allow a fair and easy comparison of the results and reproduce them at independent labs, we have chosen as black absorber a carbon cloth (product number W0S1011 from PHYCHEMi, www.phychemi.com). We have carried out systematic characterization of the carbon cloth in terms of both surface temperature in dry and wet conditions and we measured the evaporation rate under the same experimental conditions. We applied carbon cloth in the same geometry as the CoSb_x . We first investigated the asymptotic temperature achieved by carbon cloth in dry conditions under one sun illumination. The asymptotic temperature was reached by the carbon cloth sample after 7 minutes of continuous irradiation. A final temperature of 63 °C was achieved by the sample, which is far below the final temperature obtained by the CoSb_x sample (~101.7 °C). This is a rather convincing argument supporting the choice of CoSb_x not only for water evaporation, but also for other high-surface temperature applications, where high surface temperature is targeted. After having compared the maximum temperature achieved in dry conditions, we have measured the evaporation rate from the carbon cloth sample under the same experimental conditions applied for the CoSb_x system. The results of the evaporation rate are reported in the figure below and are added in the Supporting information section of the revised manuscript.

Evaporation efficiency of the membranes under 1 sun irradiation and dark condition. The evaporation rate obtained by applying a benchmarking carbon cloth is also reported.

As it is clearly visible, after 20-minute transient, the results on water evaporation from the carbon cloth ($1.08 \text{ kg m}^{-2} \text{ h}^{-1}$) are systematically lower than the ones obtained from the CoSb_x system, indicating that CoSb_x guarantees an improved mechanism of heat management during water evaporation, compared to a black, non-selective absorber.

To further elucidate the functional features of the proposed SSA contributing to the high evaporation rate, we carried out wettability tests, reported as the new Supplementary Figure 6e (also reported below). According to the wettability test (see Supplementary Fig.7e) the photothermal CoSb_3 membrane exhibits hydrophilic properties as a drop of water with a volume of $5 \mu\text{L}$ was absorbed within 696 ms.

Time-lapse snapshots of absorbing a water droplet by the photothermal CoSb_3 membranes.

We are grateful to reviewers for having suggested this control experiment, which further clarifies the originality and significance of our work.

We are also confident that the new data can solidify the conclusions of our work, obtaining a clear demonstration of the advantages offered by our new SSA as simple system for high-surface temperature and high solar water evaporation.

REVIEWERS' COMMENTS

Reviewer #1 (Remarks to the Author):

The author properly understood my comments and responded appropriately. I am satisfied with the author's corrections. Thus, I think the manuscript can be ready to be published.

ANSWERS TO REVIEWER COMMENTS (our answers in bold)

Reviewer #1 (Remarks to the Author):

The author properly understood my comments and responded appropriately. I am satisfied with the author's corrections. Thus, I think the manuscript can be ready to be published.

Answer: We are glad to read the positive answer from the Reviewer. We have now copyedited the manuscript according to the Editor's suggestion, and we look forward to obtain the final acceptance of our submission.